# Charge density waves tuned by biaxial tensile stress

A. Gallo–Frantz [1], V. L. R. Jacques [1] ✉, A. A. Sinchenko[1], D. Ghoneim [1],
L. Ortega[1], P. Godard [2], P.-O. Renault[2], A. Hadj-Azzem[3], J. E. Lorenzo[3],
P. Monceau[3], D. Thiaudière[4], P. D. Grigoriev [5,6], E. Bellec [7] & D. Le Bolloc'h [1]

The precise arrangement and nature of atoms drive electronic phase transitions in condensed matter. To explore this tenuous link, we developed a true biaxial mechanical deformation device working at cryogenic temperatures, compatible with x-ray diffraction and transport measurements, well adapted to layered samples. Here we show that a slight deformation of $TbTe_3$ can have a dramatic influence on its Charge Density Wave (CDW), with an orientational transition from $c$ to $a$ driven by the $a/c$ parameter, a tiny coexistence region near $a = c$, and without space group change. The CDW transition temperature $T_c$ displays a linear dependence with $|a/c - 1|$ while the gap saturates out of the coexistence region. This behaviour is well accounted for within a tight-binding model. Our results question the relationship between gap and $T_c$ in $RTe_3$ systems. This method opens a new route towards the study of coexisting or competing electronic orders in condensed matter.

Charge density waves (CDW) have raised considerable attention for decades due to their peculiar properties[1,2], but the interest in this phase significantly increased recently due to its competition with superconductivity in various systems[3]. As they are very sensitive to electron-phonon coupling, the application of strain is key to tuning their electronic properties. The methods to play on structural parameters are diverse: physical and chemical pressure, epitaxial strain in thin films, etc. Recently, the application of direct mechanical deformation using piezoelectric actuators[4] was successfully applied in quantum materials to drive electronic transitions. Elastoresistivity measurements performed under uniaxial stress was used to study nematic susceptibility in iron pnictides[5–8] or heavy fermion materials[9]. Although compatible with cryogenic temperatures, devices applying uniaxial stress have an intrinsic limitation in terms of flexibility, as only one direction of strain is controlled in the sample. It is, for instance, impossible to get an increase of the lattice parameters in two directions of the crystal. In addition, the true sample deformation is generally not directly measured.

Here, we present results obtained by true biaxial mechanical deformation of quasi-2D materials at cryogenic temperatures. To do so, we used a newly developed device with which tensile stress can be applied along two perpendicular axis from 80 K to 375 K (see Fig. 1b), and both electronic and structural parameters can be probed using resistivity and x-ray diffraction (XRD) measurements in the same sample. Resistivities can also be measured along and perpendicular to the deformation axis simultaneously in the same sample (see Fig. 1c). More information on the device is provided in Supplementary Information.

In this work, we focus on $TbTe_3$, one of the $RTe_3$ compounds, where $R$ is a rare-earth element of the Lanthanide family ($R$ = La, Ce, Pr, Nd, Sm, Gd, Tb, Dy, Ho, Er, Tm)[10–12]. These systems are quasi-tetragonal (see Fig. 1a), and display a highly rich phase diagram. A first CDW transition appears along $c$ in all $RTe_3$ systems below $T_c$ with wavevector $\boldsymbol{Q}_c = (0, 0, \sim \frac{5}{7} c^*)$, and a second one at lower temperature $T_{c_2}$ with wavevector $\boldsymbol{Q}_a = (\sim \frac{5}{7} a^*, 0, 0)$ for the heavier R elements ($R$ = Tb, Dy, Ho, Er and Tm)[10]. Both CDW are incommensurate with the underlying

[1]Laboratoire de Physique des Solides, Université Paris-Saclay, CNRS, 91405 Orsay Cedex, France. [2]Institut Pprime, CNRS-Université de Poitiers-ENSMA, 86962 Futuroscope-Chasseneuil Cedex, France. [3]Univ. Grenoble Alpes, CNRS, Grenoble INP, Institut Néel, 38000 Grenoble, France. [4]Synchrotron SOLEIL, L'Orme des Merisiers, 91190 Saint-Aubin, France. [5]L. D. Landau Institute for Theoretical Physics, Chernogolovka, Moscow Region 142432, Russia. [6]National University of Science and Technology 'MISiS', 119049 Moscow, Russia. [7]CEA Grenoble, IRIG, MEM, NRS, 17 rue des Martyrs, F-38000 Grenoble, France. ✉e-mail: vincent.jacques@universite-paris-saclay.fr

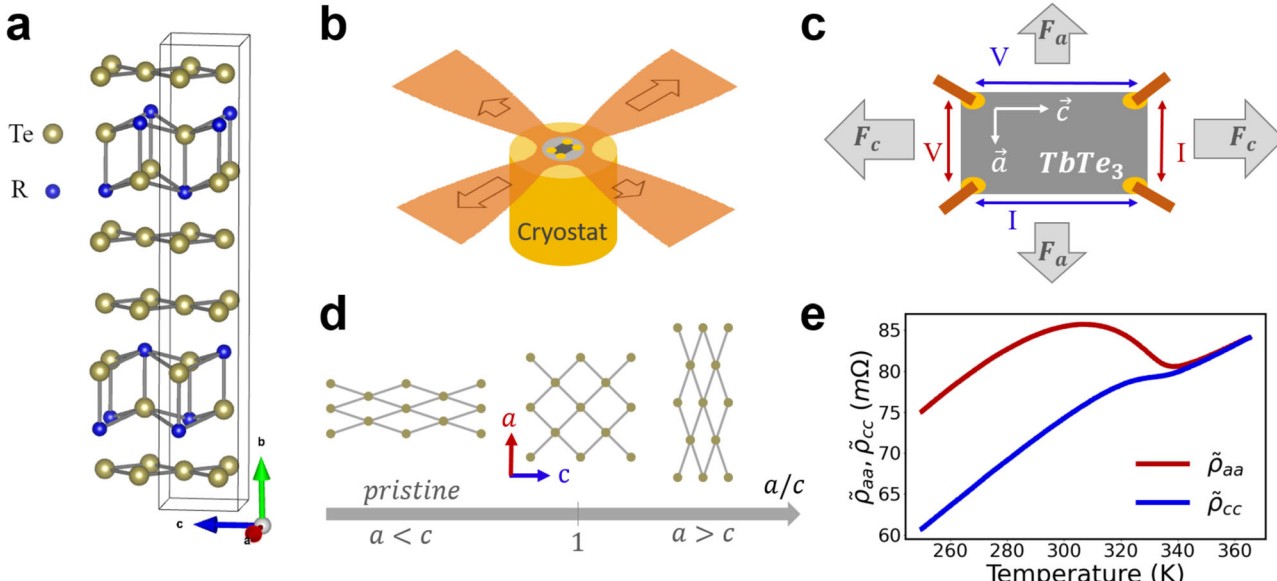

**Fig. 1 | Principle of cryogenic biaxial deformation of TbTe₃. a** The RTe₃ crystal structure is quasi-tetragonal (unit cell shown in black), and made of a succession of RTe slab intercalated by two quasi-square Te planes in the (**a**, **c**) planes. The two in-plane directions are non-equivalent due to the presence of a glide plane along **c** (space group *Cmcm*) which leads to a slight orthorhombicity ($1 - \frac{a}{c} \sim 1.3 \cdot 10^{-3}$ in TbTe₃ at 300 K[10]). **b** Schematic drawing of the biaxial tensile stress device: a thin crystalline sample is glued on a polyimide cross-shaped substrate on which tensile stresses are applied along the arms. The main in-plane directions of the crystal **a** and **c** are aligned with the arms of the cross. The bottom side of the cross is in contact with the cold finger of a nitrogen-flow cryostat. **c** Four electrical contacts are deposited at the four corners of the sample, allowing to apply currents and measure voltages along **a** or **c** directions of the crystal. The forces applied along the opposite arms of the cross have same magnitude, both along **a** and **c** (**F$_a$** and **F$_c$**, respectively). **d** Schematic view of crystal deformation as a function of a/c structural parameter. **e** Temperature dependence of $\tilde{\rho}_{aa} = \rho_{aa}/d$ and $\tilde{\rho}_{cc} = \rho_{cc}/d$ (where *d* is the sample thickness) in the pristine state of sample 1 extracted using the Montgomery technique (see Methods). They display the typical features observed in TbTe₃ compounds, i.e., a linear behavior in the normal state, above $T_c$, and a resistivity jump below $T_c \sim 337 \pm 1$ K when entering the CDW phase.

lattice period[10,13,14]. A magnetic phase appears at low temperature for the heaviest R as well as SC under pressure at ~1 K for some compounds in the series[15,16].

Recently, elastoresistivity and elastocaloric measurements performed under uniaxial stress in ErTe₃ and TmTe₃ suggested a possible CDW orientational switching from **c** to **a**[17]. A significant change of $T_{c_2}$ was also reported when the sample was deformed along **a**, while only a very slight change of $T_c$ was observed. However, in the latter study, the change of *a* and *c* lattice parameters could not be measured or changed simultaneously, and resistivity measurements were only performed along the applied stress direction, which prevents to get the full information on both CDW in all experimental conditions.

Here, we use biaxial in-plane tensile stress to independently apply controlled deformation and measure resistivities along both crystallographic axis **a** and **c** of TbTe₃ (see Fig. 1c). The resistivities obtained in the pristine state of sample 1, are presented in Fig. 1e, where $\tilde{\rho}$ indicates resistivity $\rho$ divided by sample thickness. $\tilde{\rho}_{aa}$ is particularly sensitive to the transition towards the CDW along **c** due to the gap opening at Fermi level in this direction that annihilates electronic states with velocities along **a**. This consequently induces a much larger increase of $\tilde{\rho}_{aa}$ than $\tilde{\rho}_{cc}$[18].

## Results

### Structural evolution of lattice and CDW under biaxial stress

XRD measurements were performed to follow the Bragg peaks associated with the main crystal structure during deformation (see Methods), as well as the satellite peaks associated with the CDWs along **a** and **c** (referred to as CDW$_a$ and CDW$_c$ in the following). Three non-colinear Bragg reflections (0 16 0, 0 16 1, and 1 15 0) were measured to retrieve the three lattice parameters at all forces $F_a$ and $F_c$ applied along **a** and **c**, respectively (see Fig. 2a). Their evolution is plotted as a function of $F_a$ and $-F_c$ in Fig. 2b.

*a* and *c* follow a quasi-linear behavior when applying uniaxial forces with an in-plane Poisson ratio $\nu_{ac} \sim 1$ while *b* decreases both when applying forces $F_a$ and $F_c$ with an out-of-plane Poisson ratio $\nu_{ab} \sim \nu_{bc} \sim 0.1$. This strong difference can be explained by the weak van der Waals coupling between layers along **b**. The evolution of the *a/c* ratio is plotted as a function of uniaxial forces in Fig. 2c for several temperatures. It evolves linearly as a function of applied uniaxial force, and we use this direct correlation to express all relevant quantities in terms of *a/c* in the following.

The CDW reflections can also be tracked as a function of applied force by XRD. Indeed, as CDWs add a new periodicity in the system, additional reflections appear around lattice Bragg reflections and provide important information about the static or dynamical CDW structure[19–22]. The intensity of those satellites is related to the amplitude of the periodic lattice distorsion (PLD), their position relative to the Bragg peak position gives the CDW wavevector, and their width is linked to its correlation length. Here, we measured the 1 15 ± 2/7 satellite reflections (associated with CDW$_c$) as a function of applied stress, as well as the 2/7 15 1 (associated with CDW$_a$) as it was shown to appear transiently after laser excitation[23] and suggested to appear under mechanical stress[17]. The rocking curves obtained on these satellites at T = 250 K are presented in Fig. 2d as a function of *a/c*. In the pristine state, the 1 15 ± 2/7 displays a single peak, while the 2/7 15 1 is completely absent, as expected in the absence of twin domains. When *a/c* decreases down to the lowest value reached here, the 1 15 ± 2/7 intensities increase with no clear change of width, while the 2/7 15 1 is still absent. On the contrary, when *a/c* increases, the 1 15 ± 2/7 peak intensities decrease until complete disappearance at *a/c* = 1.004. Concomitantly, the 2/7 15 1 peak increases up to ~2/3 of the maximum intensity of the 1 15 ± 2/7 with a similar width as the 1 15 2/7 (~0.5 deg). Thus, when *a/c* increases, CDW$_c$ progressively disappears while CDW$_a$ appears with

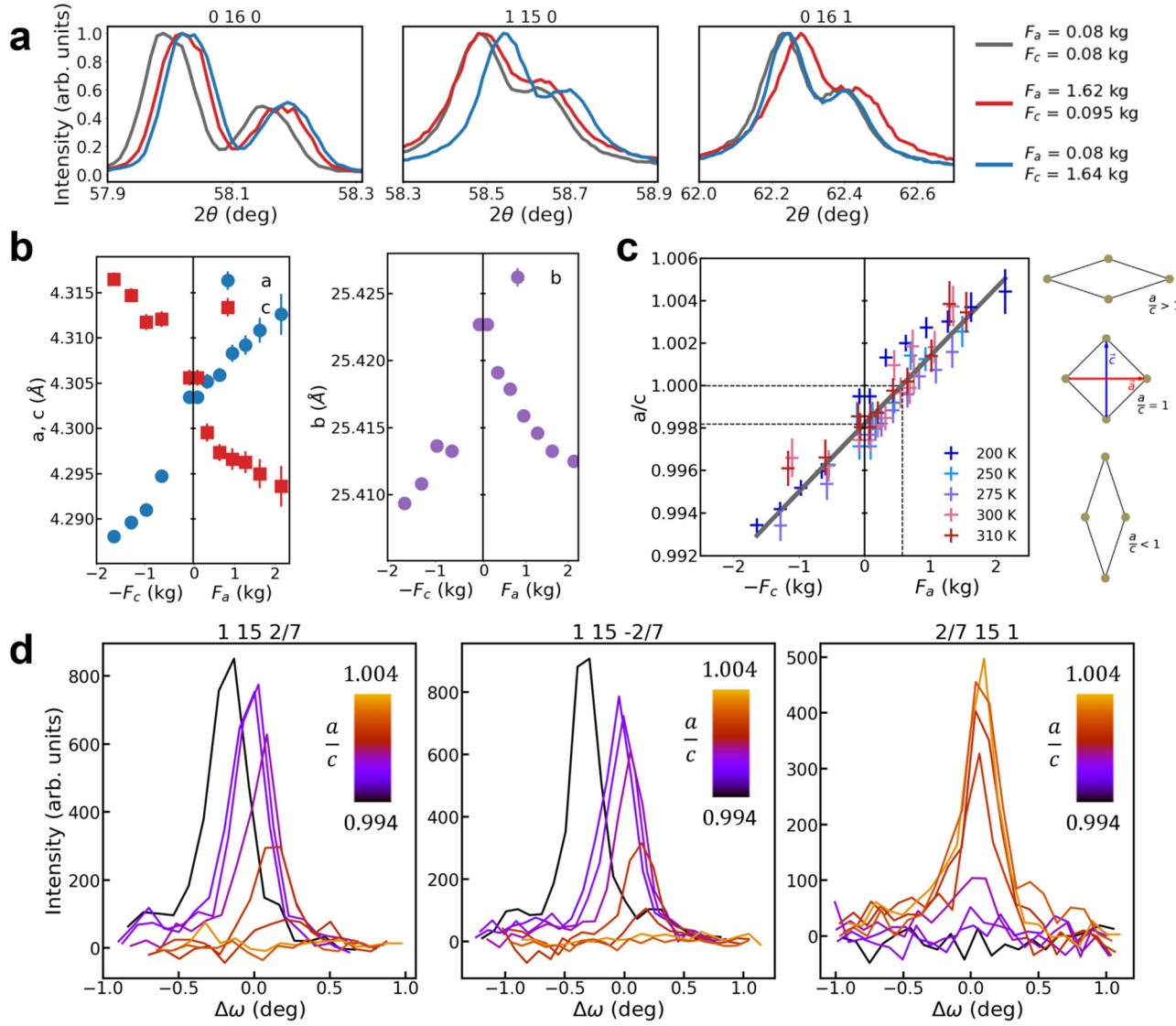

**Fig. 2 | Lattice and CDW probed by XRD under uniaxial stresses. a** 0 16 0, 1 15 0, and 0 16 1 Bragg peak intensities normalized to 1, plotted along the longitudinal $2\theta$ direction, for 3 sets of forces: without applied force (gray curves), with $F_a = 1.62$ kg (red curves) and $F_c = 1.64$ kg (blue curves). Each Bragg reflection displays the $K_{\alpha_1}$ and $K_{\alpha_2}$ components of the x-ray source. **b** Evolution of the in-plane and out-of-plane lattice constants of TbTe$_3$, obtained from the fit of the 3 non-colinear Bragg peaks shown in **a** as a function of $F_a$ and $-F_c$. $\Delta a/a \sim -\Delta c/c \sim 0.3\%$ and $\Delta b/b \sim -0.03\%$ at maximum deformation, which corresponds to an in-plane (resp. out-of-plane) Poisson ratio $\nu_{ac} \sim 1$ (resp. $\nu_{ab} = \nu_{cb} \sim 0.1$). **c** Evolution of the a/c ratio as a function of the same forces, at several temperatures between 200 K and 310 K. The gray line is

obtained by a linear fit of all data points: $a/c = \alpha F + \beta$, with $F = F_a$ for $F > 0$ and $F = -F_c$ for $F < 0$, $\alpha = 0.0032$ kg$^{-1}$ and $\beta = (a/c)_{F=0} \sim 0.9982 \pm 0.001$ which is consistent with the expected $a/c$ ratio in the pristine state. $a/c = 1$ is obtained for $F_a \sim 0.6 \pm 0.1$ kg. The error bars on the lattice constants in **b** and **c** are given by the standard deviation obtained after fitting the Bragg peaks shown **a** (see Supplementary Equation 5). The error bar on the applied force is $\pm 0.1$ kg. **d** Rocking scans on the 1 15 2/7, 1 15 –2/7 peaks associated to the CDW along **c** and 2/7 15 1 peak associated to the CDW along **a**, as a function of a/c ratio, at $T = 250$ K. The a/c ratio was computed from the uniaxial forces $F_a$ and $F_c$ using the linear fit shown in **c**. The rocking angle $\omega$ is taken relative to the peak position in the pristine state.

a similar correlation length as the previous CDW$_c$, with an intermediate coexistence region.

## Resistivities under biaxial stress

$\tilde{\rho}_{aa}$ and $\tilde{\rho}_{cc}$ were obtained at several uniaxial forces between 250 K and 375 K, well below and above $T_c$ and are plotted in Fig. 3a–c as a function of a/c for $0.993 < a/c < 1.004$.

When $a/c = 0.993$, $\tilde{\rho}_{aa}$ and $\tilde{\rho}_{cc}$ are similar to those found in the pristine state, but $T_c$ is shifted to higher temperatures. When $a/c$ increases, the resistivity jump of $\tilde{\rho}_{aa}$ (resp. $\tilde{\rho}_{cc}$) at $T_c$ decreases (resp. increases) until $\tilde{\rho}_{aa}$ is similar to the initial shape of $\tilde{\rho}_{cc}$, and inversely. In addition, all curves shift first to lower temperatures and then to higher ones. These features are well seen on the anisotropy curves $\frac{\rho_{aa}}{\rho_{cc}}$ that tend to one for $T > T_c$, and display a jump below $T_c$. This jump is

positive for low $a/c$ values, and negative for high ones. The evolution of $T_c$ is indicated by dots in the inset of Fig. 3c.

The results obtained in the case of equibiaxial deformation are highly interesting. Indeed, no clear change is observed on $\tilde{\rho}_{aa}$, $\tilde{\rho}_{cc}$ and anisotropy in this case (see Fig. 3d, e), meaning that changing the absolute values of $a$ and $c$, keeping the $a/c$ ratio globally constant does not affect the pristine CDW$_c$ state.

## A transition at $a = c$

The evolution of the main parameters measured by XRD and resistivity data are plotted as a function of the $a/c$ parameter in Fig. 4.

More specifically, in Fig. 4a, we computed the integrated intensities of the 1 15 2/7 and 2/7 15 1 satellite reflections shown in Fig. 2. As previously described, the intensity of the 1 15 2/7 decreases while 2/7 15 1

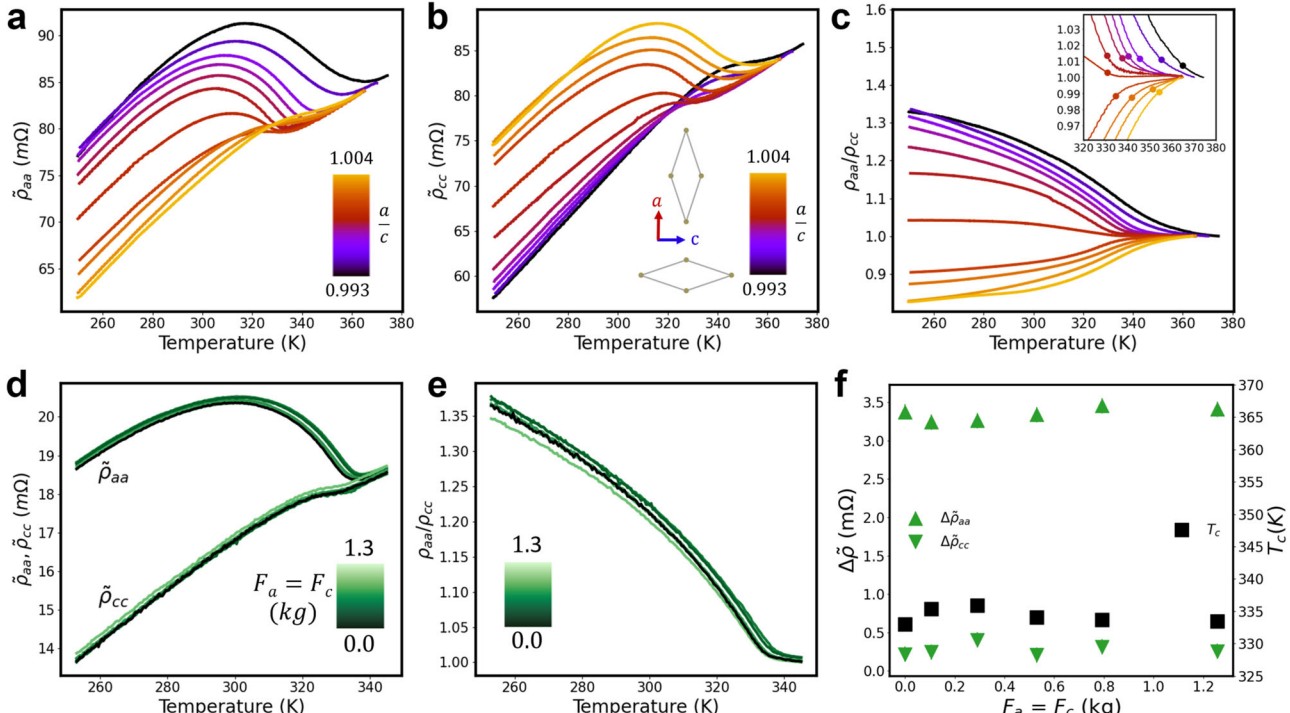

**Fig. 3 | Evolution of CDW transition under uniaxial and biaxial stress through resistivity measurements. a** $\tilde{\rho}_{aa}$ and **b** $\tilde{\rho}_{cc}$ resistivity curves obtained along $a$ and $c$, respectively for uniaxial measurements, between 250 K and 375 K, as a function of $a/c$, varied between 0.993 and 1.004 in Sample 1. **c** Anisotropy $\rho_{aa}/\rho_{cc}$ in the same temperature range, obtained from the curves shown in **a** and **b**. The dots indicate the position of $T_c$ extracted from each anisotropy curve. Inset: zoom on the anisotropy curves. **d** $\tilde{\rho}_{aa}$ and $\tilde{\rho}_{cc}$ resistivity curves obtained by equibiaxial deformation in sample 2, as a function of temperature. **e** Anisotropy curves extracted from the resistivity curves shown in **d** as a function of temperature. **f** Evolution of resistivity jumps (green triangles) and $T_c$ (black squares) for equibiaxial deformations, plotted with the same scale as uniaxial data presented in Fig. 4. No change of these parameters take place under equibiaxial deformation. Note that sample 2 behaves exactly as sample 1 under uniaxial stress, with an inversion of $\tilde{\rho}_{aa}$ and $\tilde{\rho}_{cc}$ for $F_a > 0.9$ kg (see Supplementary Information).

1 increases when $a/c$ increases from 0.994 to 1.004, following a sigmoid shape for the 2/7 15 1 and an inverse-sigmoid shape for 1 15 2/7, having the same width, and both centered at the same position $a/c = 1$ (within the error bars). The saturation value of the 2/7 15 1 is ~2/3 that of the saturation value of the 1 15 2/7, and both go down to zero when they reach their minimum value. When $a/c < 0.999$ (resp. $a/c > 1.002$), we only measure a satellite along $c$ (resp. $a$). We thus observe a continuous transformation of $CDW_c$ into $CDW_a$ when increasing the $a/c$ ratio, an intermediate coexistence phase for $0.9985 < a/c < 1.002$, and a crossing point at $a/c = 1$.

The same kind of plot has been performed for the resistivity jump $\Delta\tilde{\rho}$ obtained from $\tilde{\rho}_{aa}$ and $\tilde{\rho}_{cc}$ by taking the maximum value of $\tilde{\rho} - \tilde{\rho}_n$, where $\tilde{\rho}_n$ is the linear resistivity in the normal state, for $T > T_c$ (see Supplementary Information). The values $\Delta\tilde{\rho}_{aa}$ and $\Delta\tilde{\rho}_{cc}$ obtained with this method are shown in Fig. 4b. The shape of both curves is identical to the ones obtained for the satellite intensities in Fig. 4a: $\Delta\tilde{\rho}_{aa}$ has an inverse-sigmoid shape and $\Delta\tilde{\rho}_{cc}$ a sigmoid shape, and the same widths. They also cross around $a/c = 1$ (within error bars), with an intermediate region in the range $0.9985 < a/c < 1.0005$. Also, similarly to the satellite intensity curves, the saturation value obtained for $\Delta\tilde{\rho}_{cc}$ is ~2/3 the saturation value of $\Delta\tilde{\rho}_{aa}$. In contrast, the resistivity jumps are constant for equibiaxial deformations (see Fig. 3f).

Finally, a value of $T_c$ has been extracted from the resistivity curves by taking the second derivative of the resistivity curves and spotting the local peak corresponding to the inflection point of the resistivities (see Supplementary Information). By doing so, the $T_c$ obtained from $\tilde{\rho}_{aa}$, $\tilde{\rho}_{cc}$ and anisotropies are consistent. The evolution of $T_c$ is plotted as a function of $a/c$ in Fig. 4c. $T_c$ decreases linearly for $a/c < 1$, and then increases linearly again for $a/c > 1$, with a minimum value of 330 K found at $a/c = 1$. $T_c$ was increased by 30 K compared to the pristine

state, but could potentially go on increasing to much larger values if the $a/c$ parameter could be explored in a wider range of values. The linear dependence is consistent with the behavior reported in ref. 17, but with a much greater amplitude. Interestingly, in the case of equibiaxial deformations, $T_c$ is found to be constant within error bars (see Fig. 3f).

## Discussion

All the results presented above show that the key parameter for the CDW orientational switching from $c$ to $a$ is the structural parameter $a/c$. This result is of prime importance as it shows the direct relationship between the crystal structure and the appearance of CDWs in TbTe$_3$, and presumably in all RTe$_3$ systems. Indeed, a hint of this transition was also reported in TmTe$_3$ and ErTe$_3$ in ref. 17. Our results also show that the glide plane does not play any role in the CDW stabilization as it is still present during crystal deformation—reflections forbidden by symmetry never appeared in our experiments. When $a = c$, both $CDW_c$ and $CDW_a$ coexist, as revealed by XRD and transport measurements. Those results are extremely reproducible, both when changing the $a/c$ ratio from one phase to another, and also from one sample to another (see Supplementary Information).

The intensity of the satellite reflections associated with the CDW is generally related to the amplitude of the PLD, itself proportional to the CDW gap squared. The fact that the intensities of satellite reflections measured in XRD are indeed related to the CDW gap in TbTe$_3$ was confirmed in different experiments[15,24,25]. Here, the intensity of the satellite associated with $CDW_c$ decreases when $a/c$ increases, while the one associated with $CDW_a$ increases. The fact that we do not observe any change of width, within the resolution of the present measurement, means that if domains appear during the transition

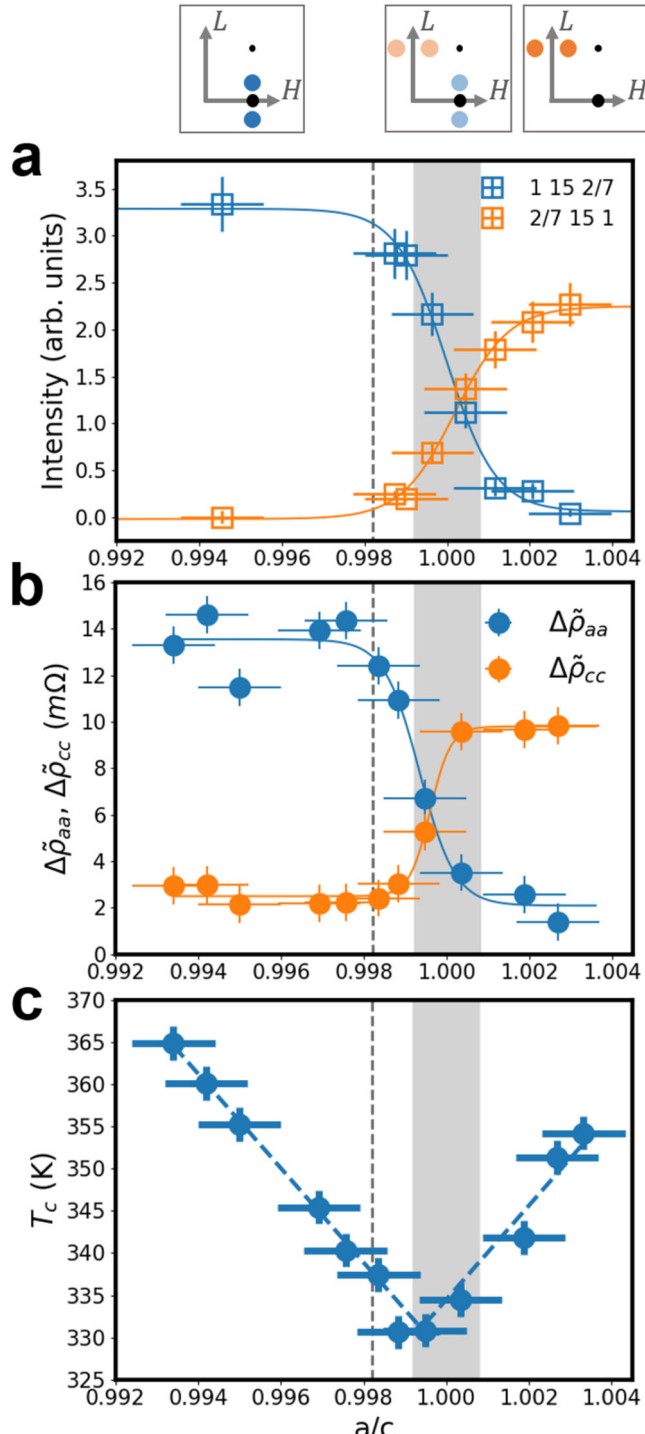

**Fig. 4 | Distinct evolution of CDW gap and $T_c$ under mechanical deformation.**
**a** Evolution of the 1 15 2/7 (blue squares) and 2/7 15 1 (orange squares) satellite intensities measured as a function of $a/c$. The blue (resp. orange) solid line is an inverse-sigmoid (resp. sigmoid) fit to the experimental points obtained on the 1 15 2/7 (resp. 2/7 15 1) satellite reflection. The three sketches in the top represent a (H 15 L) plane in reciprocal space. The big (resp. small) black dot is the 1 15 0 (resp. 1 15 1) Bragg reflection, and the blue (resp. orange) dots are the 1 15 ± 2/7 (resp. ± 2/7 15 1) reflections, with a color strength depending on the measured intensity in each $a/c$ region. The error bar on the intensity is taken as $\pm 5\sqrt{I}$ where $I$ is the satellite intensity in each experimental condition. **b** Resistivity jumps $\Delta\tilde{\rho}_{aa}$ (blue dots) and $\Delta\tilde{\rho}_{cc}$ (orange dots) extracted from the experimental curves shown in Fig. 3, plotted as a function of $a/c$ (see method in Supplementary Information). The blue (orange) solid line is an inverse-sigmoid (resp. sigmoid) fit of $\Delta\tilde{\rho}_{aa}$ (resp. $\Delta\tilde{\rho}_{cc}$). The error bar on the resistivity jumps was estimated at $\pm 0.8$ m$\Omega$ for all data points. **c** $T_c$ computed from the resistivity curve taking the average of the local peaks found in the second derivative of $\tilde{\rho}_{aa}$, $\tilde{\rho}_{cc}$ and $\rho_{aa}/\rho_{cc}$ as a function of $a/c$ (see the $T_c$ determination method in the Supplementary Information file). The error bar on $T_c$ is $\pm 2$ K (see procedure for the determination of $T_c$ in the Supplementary Information file). For all three panels, the dashed gray line is the $a/c$ value in the pristine state, and the gray filled region marks the position $a/c = 1$ with a width equal to the error bar ($\pm 0.8 \times 10^{-3}$).

from $CDW_c$ to $CDW_a$, they are never smaller than 25 nm. The observed behavior suggests more that the amplitude of the PLD along **c** vanishes while the one along **a** increases, with the same correlation length for $CDW_c$ and $CDW_a$. Consequently the gap must close along **c** and appear along **a** with a mixed state when $a = c$.

The same behavior is observed in the resistivity measurements, suggesting that the amplitude of the jumps $\Delta\tilde{\rho}_{aa}$ and $\Delta\tilde{\rho}_{cc}$ are related to the gap along **c** and **a**, respectively, and that the gap must close along **c** when $a/c$ increases while it opens along **a**, with a mixed state in the region $a = c$. The inversion of anisotropy is also an indication of this gap switching. In the pure $CDW_a$ phase (for $a/c > 1.002$), XRD and transport data consistently suggest that the gap of the $CDW_a$ phase is lower than the one in the pure $CDW_c$ phase. The gap that opens along **a**

should thus be smaller than the one along **c** in the $CDW_c$ phase. All this should appear in ARPES experiments upon tensile deformation of the sample along **a**.

The coexistence region $0.999 < a/c < 1.002$ demonstrates that a mixed state can be stabilized. It should be investigated whether they appear in separate domains in the sample, in different Te planes, or if they are superimposed in the same Te planes.

The present results can be compared to the behavior observed in hydrostatic pressure experiments or with chemical pressure obtained by changing the rare-earth element, and which were shown to induce similar effects on the CDWs[11,26]: when the lattice parameters are compressed, the CDW wavevector, the gap and $T_c$ evolve similarly in both cases. However, here, when applying tensile stresses without change of $a/c$ ratio in the equibiaxial experiment, the resistivity curves do not change. We conclude that the gap and $T_c$ do not change in the negative pressure region. To observe variations of gap and $T_c$, the parameter $a/c$ has to change significantly. In the experiments reported in ref. 26, the parameter $a/c$ changes and tends to 1 when pressure is applied (above 3GPa in CeTe$_3$). In these conditions, $T_c$ decreases, which is in qualitative agreement with our observations: $T_c$ is minimum when $a/c = 1$. Quantitatively, the variation of $T_c$ reported in ref. 26 is much larger than reported here. Concerning the evolution of the gap, they also report vanishing CDW satellite intensities, suggesting a disappearance of the gap at high pressures, while $a/c = 1$. Here, no change is observed in equibiaxial deformation, but the experiments should be repeated in equibiaxial conditions while $a/c = 1$. The role of the relative variation of the $b$ parameter should also be investigated further to compare our experiment to hydrostatic pressure measurements.

The behavior observed here is very similar to the one observed after fs laser excitation reported in ref. 23, with the CDW appearing along **a** while the one along **c** disappears. Contrary to the equilibrium CDWs at low temperatures in some RTe$_3$ compounds, the CDWs appearing along **a** and **c** in this reference and in our study have only a tiny coexistence region. This makes the new $CDW_a$ unique and different from the equilibrium coexistence of $CDW_c$ and $CDW_a$ at low temperatures. Following the results presented in ref. 23, the lattice parameters of LaTe$_3$ after laser pulse excitation do not vary by more than 0.02% when the transient $CDW_a$ appears, which indicates that the driving mechanism is different from the one we report here. In our case, both CDWs are stable as long as the $a/c$ parameter is kept constant.

Finally, the most striking feature is the evolution of $T_c$ compared to the evolution of the gap, as extracted from XRD and resistivity measurements. Both satellite intensities and $\Delta\tilde{\rho}$ saturate when

$a/c < 0.9985$ and $a/c > 1.002$, and therefore, the gap values saturate in this region. However, $T_c$ keeps diverging linearly, which questions its relation with the gap. The linear dependence of $T_c$ with respect to $a/c$ can be understood within the tight-binding model, with good quantitative agreement (see Methods). The model predicts an increase of $T_c$ by $\delta T_c \approx 26K$ for $\delta a/a \approx 0.2\%$, which is comparable to the experimental values reported here. The same applies to deformations along $c$, and explains the linear dependence of $T_c$ as a function of deformation. The system actually chooses the CDW along the axis with the highest $T_c$. For $a > c$, $T_c$ is higher for $CDW_a$, while for $a < c$, $T_c$ is higher for $CDW_c$. At $a = c$, the transition temperatures are equal, and the system is degenerate.

In this article, we report in-situ XRD and transport to follow the structural and electronic properties of the CDWs in $TbTe_3$ during biaxial mechanical deformation between 200 K and 375 K. By directly measuring both CDW satellite intensities and resistivities along $a$ and $c$, we show that the CDW orientation continuously flips from $c$ to $a$ when the $a/c$ ratio is increased, with a coexistence of both orientations when $a = c$. As those two parameters are linked to the gap, we can infer that the gap position changes from one band to another, and that both bands are gapped when $a = c$. Moreover, the transition temperature displays a linear behavior with respect to $a/c$, with a minimum at $a = c$, which is well accounted for by a 2D tight-binding model in the ($a$, $c$) Te planes, with good quantitative match between theory and experiment. For the pure $CDW_a$ and $CDW_c$ phases, the gap saturates while $T_c$ does not, which questions the relationship between those quantities in $RTe_3$ systems. Our study opens new perspectives for the exploration of many electronic phase transitions in condensed matter systems with application of biaxial tensile stress at cryogenic temperatures.

## Methods

### Samples
The $TbTe_3$ samples were grown by the self-flux method, as described in ref. 13. All samples were cut to have an in-plane rectangular shape, and were mechanically exfoliated down to few μm thickness to get homogeneous deformation in the volume. Results obtained on two different $TbTe_3$ single-crystals are presented in the main text. Sample 1 was probed both by XRD and transport to get the full structural and electronic properties, with uniaxial deformation, and sample 2 was probed by transport under uniaxial and biaxial deformation. The clearly different behavior of $\rho_{aa}$ and $\rho_{cc}$ allowed us to directly determine the crystal orientations and $T_c$ (see Supplementary Information)[18]. Other samples were measured by XRD or transport and give the same results, as presented in Supplementary Information. Sample 1 (resp. sample 2) has a 1:1.17 (resp. 1:2) in-plane aspect ratio, and 2.5 μm (resp. 10 μm) thickness.

### Biaxial tensile stress device
The principle of the new device developed here is similar to the one described in ref. 27, with the sample glued on a deformable substrate that is mechanically stretched, and adapted to host a nitrogen-flow cryostat to reach temperatures in the range 80–375 K. The samples are glued at the center of a 125 μm-thick polyimide cross-shaped substrate, with $a$ and $c$ in-plane directions aligned with the arms of the polyimide cross. The four arms of this substrate are attached to four independent motors that can pull on each branch separately. The forces applied along the four arms are measured using calibrated force gauges, and are given in kg. The center of the cross is covered with a thin gold layer on its bottom surface and lies on the cold finger of a Konti-Micro cryostat from CryoVac GmbH, allowing to reach temperatures in the range 80–375 K. Apiezon grease is used to get a good thermal transfer between the cold finger and the polyimide cross. In practice, the same motor displacement is used for opposite arms to keep the sample at the same position in the device. This whole setup is enclosed in a vacuum chamber for cryogenic operation, and allows access for incoming and outgoing x-rays to perform XRD in reflection geometry through a 300 μm-thick Polyether-ether-ketone dome. Four wires are also available in the cryostat for 4-point transport measurements.

### XRD measurements
The experiment shown in the main text were performed at LPS with a 8 keV x-ray beam generated by a Cu rotating anode source (Rigaku RU-300B), equipped with a multilayer monochromator suppressing the $K_\beta$ emission line but allowing the $K_{\alpha_1}$ and $K_{\alpha_2}$ ones. The sample mounted in the biaxial tensile stress device was positioned at the center of rotation of a Huber Eulerian 4-circle diffractometer to perform wide-angle XRD, and detection was performed with a 2D detector (Timepix from ASI) located 82 cm after the sample. The 3 Bragg reflections (0 16 0, 1 15 0, and 0 16 1) recorded on the 2D detector were projected along the $2\theta$ direction of reciprocal space to compute the three lattice parameters $a$, $b$, and $c$ for each set of applied forces (see Supplementary Information). The absence of twin domains was demonstrated by checking that the forbidden 0 15 1 reflection was indeed not measurable.

### Transport measurements
Four contacts were deposited at the four corners of the crystal, and the Montgomery method[28–31] was used to get both $\tilde{\rho}_{aa}$ and $\tilde{\rho}_{cc}$ resistivities along the $a$ and $c$ respectively by applying a fixed current with a Keithley 2611 Sourcemeter, and measuring the voltage with a Keithley 2182a Nanovoltmeter. For each measurement, the forces $F_a$ and $F_c$ were changed at 375 K, and the resistances were measured during cooling down to 250 K at a fixed rate of 1.5 K/min. An electronic device allowed us to switch $I^+$ and $V^-$ every second to get $R_{aa}$ and $R_{cc}$ during the same temperature ramp. The ratio of transverse dimensions of the sample was obtained by fixing the anisotropy $\rho_{aa}/\rho_{cc}$ to $c^2/a^2$ in the normal state and used to compute the resistivities normalized by sample thickness $d$: $\tilde{\rho}_{aa} = \rho_{aa}/d$ and $\tilde{\rho}_{cc} = \rho_{cc}/d$. See Supplementary Information for detailed formalism. Resistivity measurements presented in Fig. 3a–c were performed in Sample 1 after measuring the XRD data, and the ones in Fig. 3d–f in Sample 2. See Supplementary Information for complete resistivity measurements in Sample 2.

### Tight-binding model
In this description, the electron dispersion of the two bands formed by the $p_x$ and $p_z$ Te orbitals $\varepsilon_\pm$ keeps its form (where $x$ and $z$ are along $a$ and $c$ respectively):

$$\varepsilon_\pm = -2t_\parallel \cos\left(k_x^* \pm k_z^*\right) - 2t_\perp \cos\left(k_x^* \mp k_z^*\right) \qquad (1)$$

where $k_x^* = k_x a/2$ and $k_z^* = k_z c/2$, and $t_\parallel$ and $t_\perp$ are the transfer integrals parallel and perpendicular to the orbital direction. Both lattice parameters and transfer integrals depend on the applied stress. At $a = c$, Eq. (1) coincides with the accepted tight-binding dispersion given by Eq.(1) in ref. 32. The CDW transition temperature is given by the condition :

$$U(\boldsymbol{Q_0})\chi(T_c, \boldsymbol{Q_0}) = 1 \qquad (2)$$

where the static electron-electron interaction $U(\boldsymbol{Q})$ includes both Coulomb and phonon-mediated interaction, and the Lindhard susceptibility reads:

$$\chi(T_c, \boldsymbol{Q}) = \sum_{\alpha,\alpha'} \sum_{k_x, k_z} 16 \frac{n_F(E_{\boldsymbol{k},\alpha}) - n_F(E_{\boldsymbol{k}+\boldsymbol{Q},\alpha'})}{E_{\boldsymbol{k}+\boldsymbol{Q},\alpha'} - E_{\boldsymbol{k},\alpha}} \qquad (3)$$

where $n_F(\varepsilon) = 1/(1 + exp[(\varepsilon - E_F)/T])$ is the Fermi-Dirac distribution function, $\alpha, \alpha' = \pm$ label the subbands and $E_{\boldsymbol{k},\alpha}$ differs from Eq (1) only near the intersection points of two bands in momentum space, as given by Eq. (2) in ref. 33. As $\varepsilon_\pm$ and $E_{\boldsymbol{k},\pm}$ depend only on $k_x^*$ and $k_z^*$, and

as $\chi(T_c, \mathbf{Q})$ has a $x \leftrightarrow z$ symmetry, an equibiaxial stress does not affect the Lindhard susceptibility. The main difference in the transition temperatures of the CDW along $\mathbf{a}$ and $\mathbf{c}$ thus comes from the difference of $U(\mathbf{Q})$ when $a \neq c$. The electron-electron coupling mainly comes from the Coulomb interaction, screened by conducting electrons: $U(r) = e^2 exp(-\zeta r)/r$, where the inverse Debye screening radius $\zeta = \sqrt{4\pi e^2 \rho_F}$ and $\rho_F$ is the density of states at the Fermi level. In TbTe$_3$, $\zeta \approx a^{-1} \approx (4.3)^{-1}$ Å$^{-1}$. The Fourier transform of the screened Coulomb potential is :

$$U(\mathbf{Q}) \approx \frac{4\pi e^2}{\mathbf{Q}^2 + \zeta^2} \tag{4}$$

The CDW wavevector $\mathbf{Q_0}$ is given by the maximum of the Lindhard susceptibility, Eq. (3), and gives a fixed product $Q_{0x}a = Q_{0z}c \approx 10\pi/7$, when $a = c$. An increase of the lattice constant $a$ decreases the CDW wavevector $Q_{0x}$ and increases the CDW coupling $U(\mathbf{Q_0})$ according to Eq. (4). An increase of $a$ by $\delta a/a = 0.1\%$ results in a decrease of $Q_{0x}^2$ by 0.2%. Since $\zeta << Q_{0x}$, according to Eq. (4), this gives an increase of $U(\mathbf{Q_0})$ by 0.2%. According to Eq. (2), an increase $\delta U$ of $U(\mathbf{Q_0})$ raises the CDW transition temperature $T_c$ by $\delta T_c$ given by Eq. (2):

$$\frac{\delta U}{U} = \frac{-\delta \chi}{\chi} = \frac{d\chi(T, \mathbf{Q_0})}{dT} \frac{-\delta T_c}{\chi(T, \mathbf{Q_0})} \tag{5}$$

because the electron susceptibility $\chi(T, \mathbf{Q_0})$, approximately given by Eq. (3) decreases with the increase of $T$. The temperature dependence of the Lindhard susceptibility was calculated in ref. 33 for the second CDW in ErTe$_3$ and in ref. 34 for the Q-dependence of the first CDW for various parameters of electron dispersion. To find $d\chi/dT$ for the first CDW in relevant temperature range and for the transfer integrals $t_\parallel \approx 2$ eV, $t_\perp \approx 0.37$ eV and the Fermi energy $E_F \approx 1.48$ eV in TbTe$_3$[32], we performed new calculations of $\chi(T, \mathbf{Q_0})$ using Eq. (3) (see Supplementary Information), from which the slope of the temperature-dependent susceptibility can be extracted :

$$\eta = \frac{d[ln\chi(T, \mathbf{Q_0})]}{dT} = \chi^{-1} \frac{d\chi}{dT} \approx -1.5 \cdot 10^{-4} K^{-1} \tag{6}$$

We thus can compute $\delta T_c$ for an increase $\delta a/a \approx 0.2\%$, comparable to the experiment described here:

$$\delta T_c = \frac{\delta U(\mathbf{Q_0})}{\eta U(\mathbf{Q_0})} \approx -\frac{2\delta a}{a\eta} \approx \frac{4 \cdot 10^{-3}}{1.5 \cdot 10^{-4}} \approx 26 K \tag{7}$$

## Data availability
The data and associated codes used to reproduce the figure of the main manuscript are available at https://doi.org/10.6084/m9.figshare.25400065. Raw data are available from the corresponding authors upon request.

## Code availability
The codes used to produce the figures of the main article are available at https://doi.org/10.6084/m9.figshare.25400065. The other codes used for raw data treatment are available from the corresponding authors upon request.

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

## Acknowledgements

A.G.-F., A.A.S., L.O., V.L.R.J, D.L.B., P.Go., P.-O.R., P.Gr. A.H.-A., and P.M. are supported by ANR-RSF Grant no. ANR-21-CE30-0055 "BISCEPS-QM" and RSF-22-42-09018. J.-E. L. and P.M.'s work is supported by the ANR-DFG Grant no. ANR-18-CE92-0014-03 "Aperiodic". All authors acknowledge Synchrotron SOLEIL for providing beamtime. A.G.-F., V.L.R.J., and D.L.B. acknowledge S. Cabaret and V. Klein for device conception and S. Rouzières and P. Joly for experimental support.

## Author contributions

A.H.-A., J.E.L., and P.M. grew and provided the TbTe$_3$ samples. A.A.S. and A.G.-F. prepared the samples for transport and x-ray experiments. A.G.-F. and V.L.R.J. performed and analyzed the laboratory XRD experiments. A.G.-F., A.A.S. and V.L.R.J. performed and analyzed the transport measurements. A.G.-F., A.A.S., D.G., L.O., V.L.R.J., D.L.B., P.Go., P.-O.R., D.T. and E.B. participated in the experiments at synchrotron SOLEIL. P.Gr. developed the theoretical part. V.L.R.J. wrote the initial draft, and all authors participated in the discussion and correction of the manuscript. V.L.R.J. and D.L.B. led the project.

## Competing interests

The authors declare no competing interests.
