## [Peer Review File · Nature Communications]

REVIEWER COMMENTS

Reviewer #1 (Remarks to the Author):

This study investigates the effect of tensile stress on CDW in TbTe₃ under biaxial tensile, which enriches the uniaxial stress study reported by JAW Straquadine et al. (Phys. Rev. X 12, 021046). The manuscript demonstrates that the orientation of the CDW continuously transitions from the c-axis to the a-axis as the a/c ratio increases, showing a coexistence of both orientations when a = c. The study also provides additional insight into the non-equilibrium CDW states reported by F. Zhou et al. (Nature Communications volume 12, 566 (2021)) and A. Kogar et al. (Nature Physics volume 16, 159–163 (2020)). These studies had similarly observed competition between the CDWs along a- and c-axes. This study suggests that the driving mechanisms for this competition under biaxial stress and fs-pulses are different. Additionally, the technique used to apply biaxial stress in this study is versatile and could serve as a valuable example for future research on various material systems in the field and related fields.

The overall quality of the work is good, and the results are clearly presented. I recommend the publication of this manuscript upon the following revisions:

- The cross-shaped substrate could experience shear stress in certain areas when stretched. How has the uniformity of the tensile stress across the sample been confirmed?
- Fig 2b suggests that applying the same amount of force along the a- and c-axes does not necessarily result in equivalent extension of the lattice parameters. Therefore, the assumption in Fig 3d-f that $F_a = F_c$ always maintains $a = c$ should be verified.
- In the discussion section, it is stated that no change is observed in equibiaxial deformation. However, The T_c of ErTe₃ ($T_c = 250$ K) with $a = c$ (smaller lattice constants than those of TbTe₃) reported previously by JAW Straquadine is significantly lower than TbTe₃ when $a = c$. It would strengthen the manuscript if a qualitative explanation were provided for both the difference in T_c for different lanthanides and the reason for the independent T_c under equibiaxial deformation.

Reviewer #2 (Remarks to the Author):

In the manuscript by Gallo-Frantz et al., the authors create a bi-axial strain device and measure the x-ray diffraction and resistivity of TbTe₃ under strains with different a/c ratios to further understand the CDW mechanism in the material. The authors are able to demonstrate switching of the CDW orientation and find that the critical parameter for the switching is the a/c structural parameter. The diffraction and resistivity data are clear and the systematic manipulation of the CDW is evident. The data well supports the authors' conclusions. However, the conclusions seem to corroborate previous investigations into the RTe₃ family of materials and significant new insights regarding the CDW mechanism are lacking. It is clear the authors spent effort to create a bi-axial strain device to make these measurements possible, but no information regarding their instrument developments are included in this manuscript and the reference points towards a future manuscript. Hence, the new information, new instrument developments, and new insights into the CDW mechanism are limited and while the results should be published in some format, they are not at the impact level required for publication in Nature Communications. Below are more points to consider.

Given the body of work on this material, the results regarding the CDW do not seem surprising and are even expected. As the authors point out, the effects of chemical pressure on the lattice and resulting changes to the CDW have been outlined in Refs. 11 and 31 and evidence for CDW orientation switching has been observed in Ref. 17. While the present work provides new systematic investigation of the connection between lattice parameter and CDW order, it simply seems to confirm previous results. If there was a significant effort to create a bi-axial strain device to make such measurements possible and if this device can be used on a wide variety of material investigations, then the impact of the manuscript would increase, but these details are reserved for a future publication.

There is a lengthy discussion connecting the diffraction intensity and electronic gap. While the discussion is likely true, there are no direct measurements of the gap. There is discussion of ARPES measurements to confirm the behavior, but the discussion is speculative and should be acknowledged as such. To make a connection between the current results and the electronic structure, the authors point to Ref. 23, which looks at LaTe₃. However, there are other references, e.g. Science 321, 1649 (2008) and Phys. Rev. B 93, 024304 (2016), that investigates TbTe₃ under femtosecond laser excitation. It seems more direct connections between the lattice and electronic structures, and the deformation potential could be made by comparing with results from laser excitations on the same material.

On a clerical note, the order of the references should be checked, e.g., I see a pointer to Ref. 12 before Ref. 11 in the main text.

Reviewer #3 (Remarks to the Author):

Gallo-Frantz et al. present a comprehensive study of stress-tuned charge density wave (CDW) transitions in rare earth tritellurides (RTe₃) using novel techniques involving x-ray diffraction and transport measurements under bi-axial stress at cryogenic temperatures. The paper is well-structured and convincingly presents a set of new data revealing the behavior of CDW in RTe₃ under both symmetry breaking ($F_a \neq F_c$) and symmetry preserving ($F_a = F_c$) bi-axial stress. This paper is suitable for publication after the authors address the following questions and suggestions.

1. **Role of Lattice Parameters:** In Figure 3(d-f), the authors observed that the CDW state and transition temperature are barely affected by symmetry preserving biaxial stress. This observation contrasts with previous reports, such as PhysRevB.77.035114, which found that the CDW transition changes with chemical pressure. If the in-plane a and c lattice parameters do not play a key role, as suggested by the authors, can the authors provide their insights and discussions on what instead determines the CDW transition temperature? For example, as the symmetry preserving biaxial stress is applied, how does the out-of-plane lattice parameter b change?
2. **Clarification of CDW Domains:** In lines 247 to 250, the authors concluded that they did not observe any change in the width of the Bragg peaks, which does not support the appearance of domains during the transition from CDW_c to CDW_a. However, in preprint arXiv:2306.14755, Fig. 2(c), a clear discontinuity in the a-axis lattice parameter was observed, suggesting the coexistence of CDW_a and CDW_c domains. Additionally, the width of Bragg peaks in the preprint displayed a finite dependence on strain. Can the authors comment on the discrepancy between their results and those presented in arXiv:2306.14755?
3. **Symmetry of TCDW vs. a/c:** Due to the existence of a glide plane, the system does not retain higher symmetry even at $a = c$. Therefore, there is no apparent reason for TCDW vs. a/c to be symmetric with respect to $a = c$. Nevertheless, figure 4(c) demonstrates that TCDW has a linear dependency on a/c and is almost symmetric with respect to $a = c$. The author's argument is that the glide plane plays no role in the CDW formation. However, x-ray diffraction data in Fig. 4(a) shows that the CDW_a and CDW_c states saturate to different maximum intensities, indicating that they are not identical, hence the CDW intensity is asymmetric with respect to $a = c$. The authors should provide an explanation for the inconsistency between these two sets of data.
4. **Tight-Binding Model:** the authors used a tight-binding model to explain the linear dependence of TCDW on a/c. Does the model also predict the lack of change in TCDW under biaxial stress? If not, the authors should elaborate on why the tight-binding model is not applicable in the condition of bi-axial stress.
5. **Technical Details:** Is the experimental setup capable of applying compression stress? If not, why?

Reviewer #4 (Remarks to the Author):

REPLY TO REVIEWERS' COMMENTS

Reviewer #1 (Remarks to the Author):

This study investigates the effect of tensile stress on CDW in TbTe₃ under biaxial tensile, which enriches the uniaxial stress study reported by JAW Straquadine et al. (Phys. Rev. X 12, 021046). The manuscript demonstrates that the orientation of the CDW continuously transitions from the c-axis to the a-axis as the a/c ratio increases, showing a coexistence of both orientations when a = c. The study also provides additional insight into the non-equilibrium CDW states reported by F. Zhou et al. (Nature Communications volume 12, 566 (2021)) and A. Kogar et al. (Nature Physics volume 16, 159–163 (2020)). These studies had similarly observed competition between the CDWs along a- and c-axes. This study suggests that the driving mechanisms for this competition under biaxial stress and fs-pulses are different. Additionally, the technique used to apply biaxial stress in this study is versatile and could serve as a valuable example for future research on various material systems in the field and related fields.

The overall quality of the work is good, and the results are clearly presented. I recommend the publication of this manuscript upon the following revisions:

The authors would like to acknowledge Reviewer #1 for his/her careful reading of the manuscript and his/her relevant comments. Our answers to his/her questions are listed below.

- The cross-shaped substrate could experience shear stress in certain areas when stretched. How has the uniformity of the tensile stress across the sample been confirmed?

This is a good point indeed, the kapton substrate can experience shear deformation in some areas, but it is negligible in the 2x2mm² area on which the sample has been glued. The shear deformation was estimated at each step of the experimental work: first theoretically using finite element calculations, then experimentally by Digital Image Correlation (DIC) measurements on the bare substrate and finally directly measured on the sample by XRD during the in-situ experiment under stress. All these measurements will be part of an instrumental paper detailing the cryogenic biaxial tensile stress device performances. However, as pointed out by Reviewer #1, it is important to show here that the sample does not experience shear deformation, so we provide some more details on the device and its performances in the Supplementary Material (section I.B), in the conditions corresponding to the data shown in the main text.

As shown in section I.B.1. of the Supplementary Material, the DIC measurements performed in uniaxial mode show a fairly linear behaviour of ε_{11} and ε_{22} along the two orthogonal tensile stress axis of the device, up to 2kg, with very low shear strain ($|\varepsilon_{12}| < 0.04\%$). In equibiaxial mode, the measurements performed on the bare substrate show that the deformation is equivalent in the two orthogonal stretching directions: $\varepsilon_{11} \sim \varepsilon_{22}$ up to $F_a = F_c = 2\text{kg}$ (i.e. more than the maximum force applied in equibiaxial mode in the manuscript), with a shear $|\varepsilon_{12}| < 0.06\%$. The sample glued on this substrate should thus follow the same deformations, without shear component.

To check this directly on the sample, we computed the shear component extracted from the 3 non-collinear Bragg reflections (same data as the ones presented in the manuscript, sample 1). This analysis is described in section I.B.2. of the Supplementary Material. Finite shear components should induce a distortion of the unit cell from an orthorhombic to a triclinic one. Thus any shear stress would have

resulted in rotational components of the measured wavevectors associated to the set of Bragg reflection that we measured and reported in Fig. 2. We computed the three reciprocal angles from the three Bragg reflections, and find that they are all equal and bound around 90° with a standard deviation less than 0.15° , with no dependence on applied uniaxial force. We thus neglected shear deformation in first approximation.

The uniformity of the tensile stress across the sample was also checked by both DIC measurements and XRD. The DIC images shown in Suppl. Fig. 2c show that the central zone of the substrate, where the sample is glued, has homogeneous deformation over more than $2 \times 2 \text{ mm}^2$, larger than the sample surface. In addition, the Bragg reflections obtained by XRD with a $1 \times 1 \text{ mm}$ x-ray beam show a solid displacement of the Bragg peaks, without broadening (see Fig. 2a of main text for Sample 1 and Fig. 6 of Supplementary Materials for Sample 2), which is in agreement with a homogeneous deformation of the sample at the millimetre scale.

- Fig 2b suggests that applying the same amount of force along the a- and c-axes does not necessarily result in equivalent extension of the lattice parameters. Therefore, the assumption in Fig 3d-f that $F_a = F_c$ always maintains $a = c$ should be verified.

Fig 2b indeed shows that we can have slightly different behaviours along the two perpendicular axis when we perform uniaxial deformations along a and c . However, the experimental parameter a/c is extracted at each point from its real measured value, which avoids any issue related to non-equivalent deformations in the experimental setup.

For the equibiaxial measurements presented in the manuscript, we only measured the resistivities in this sample and did not follow the lattice parameters by XRD. Based on the device characterization, we thus assumed a and c would evolve accordingly, with an increase of a and c by the same quantity.

As Reviewer #1 suggests, we therefore performed additional XRD measurements on 4 new samples to check the evolution of their lattice parameters during equibiaxial deformation (at room temperature, in the same conditions as described in the manuscript). The evolution of the three lattice parameters a , b and c are shown in the figure below, as well as the a/c ratio, as a function of the applied force.

These measurements show approximately the same features in all samples: a increases by $\sim 0.1\%/kg$ whereas c does not increase much in the range of forces we explored. b decreases by around $0.01\%/kg$, i.e. 10 times less than the in-plane deformation, which corresponds to the data obtained for uniaxial deformations presented in the manuscript. The small discrepancies between samples could be attributed to different sample thicknesses. The most important feature however is the a/c parameter, that does not significantly vary up to the maximum force applied in the data of the manuscript ($F_a = F_c = 1.3kg$). The a/c value changes by only 0.0015 around the average value.

These observations do not question the main message of the article: the relevant parameter to explain the changes of $\Delta\rho$ and T_c is the a/c ratio. Here, even in these conditions in equibiaxial mode for which a and c do not have exactly the same expansion, the a/c ratio is fairly constant in the range of applied forces, and this explains the behaviour shown in Fig. 3f.

The interesting and unexpected variations of a and c as a function of equibiaxial force should be explored further to understand why they tend to get closer to $a = c$. This is a similar behaviour as the one reported by Sachetti et al. (PRB 79, 201101 (2009)) as a function of hydrostatic pressure, for which a and c become equal above 3GPa.

Modifications have been done in the text to correct all sentences referring to a and c increasing by the same amount in equibiaxial mode, and replaced by sentences referring to the much smaller variation of a/c ratio in equibiaxial mode than in uniaxial along the a-axis.

- In the discussion section, it is stated that no change is observed in equibiaxial deformation. However, The T_c of ErTe₃ ($T_c = 250$ K) with $a = c$ (smaller lattice constants than those of TbTe₃) reported previously by JAW Straquadine is significantly lower than TbTe₃ when $a = c$. It would strengthen the manuscript if a qualitative explanation were provided for both the difference in T_c for different lanthanides and the reason for the independent T_c under equibiaxial deformation.

Indeed, the measurements reported by Straquadine et al. in TmTe₃ and ErTe₃ show significantly lower transition temperatures when $a = c$ than the one reported in our work in TbTe₃. If a/c was the only parameter fixing T_c in all RTe₃ systems, then we could indeed expect to find the same T_c for all components when $a = c$. However, this is not the case. The effect of the stress applied in the (a, c) plane of RTe₃ systems results in a change of T_c relative to the intrinsic T_c of the corresponding material. The tight-binding model developed in the Methods section of our manuscript provides a calculation of this expected relative change ΔT_c when the lattice is distorted from square Te planes ($a = c$), starting from the specific T_c given by other parameters of the system under consideration. The balance between electron energy gain (due to the gap opening) and entropy (due to disorder) gives T_c , but RTe₃ systems have different gap values and electron densities near Fermi level, as well as different $t_{//}$ and t_{\perp} transfer integrals, which induce different conditions on the Lindhard susceptibility and Coulomb interactions, and thus different intrinsic T_c , even if $a = c$.

The general message here is thus that for a specific system - TbTe₃ here - T_c varies linearly with a/c ratio, starting from different absolute values of T_c , that are intrinsic to each RTe₃ system.

Reviewer #2 (Remarks to the Author):

In the manuscript by Gallo-Frantz et al., the authors create a bi-axial strain device and measure the x-ray diffraction and resistivity of TbTe₃ under strains with different a/c ratios to further understand the CDW mechanism in the material. The authors are able to demonstrate switching of the CDW orientation and find that the critical parameter for the switching is the a/c structural parameter. The diffraction and resistivity data are clear and the systematic manipulation of the CDW is evident. The data well supports the authors' conclusions. However, the conclusions seem to corroborate previous investigations into the RTe₃ family of materials and significant new insights regarding the CDW mechanism are lacking. It is clear the authors spent effort to create a bi-axial strain device to make these measurements possible, but no information regarding their instrument developments are included in this manuscript and the reference points towards a future manuscript. Hence, the new information, new instrument developments, and new insights into the CDW mechanism are limited and while the results should be published in some format, they are not at the impact level required for publication in Nature Communications. Below are more points to consider.

Given the body of work on this material, the results regarding the CDW do not seem surprising and are even expected. As the authors point out, the effects of chemical pressure on the lattice and resulting changes to the CDW have been outlined in Refs. 11 and 31 and evidence for CDW orientation switching has been observed in Ref. 17. While the present work provides new systematic investigation of the connection between lattice parameter and CDW order, it simply seems to confirm previous results. If there was a significant effort to create a bi-axial strain device to make such measurements possible and if this device can be used on a wide variety of material investigations, then the impact of the manuscript would increase, but these details are reserved for a future publication.

The authors would like to thank Reviewer #2 for his/her careful reading of the manuscript, and for pointing out the quality of the presented data. We also acknowledge him for his/her comments. In the following, we address all his/her concerns point by point.

Reviewer #2's first concern is that, in his view, the results presented in the manuscript merely corroborate previous results and that there is nothing significantly new regarding CDW mechanism. We strongly disagree with this assessment. In this paper, our results go well beyond the already-observed elastoresistivity data performed under uniaxial deformation that suggested the appearance of a CDW along the a-axis in a parent compound [Straquadine et al. PRX 2023]. However, the CDW transition is both an electronic *and* a structural transition, so the combination of resistivity and XRD measurements are essential to unambiguously prove the appearance of a new CDW state in these materials. We carry out this simultaneous measurement in this paper, which had never been done before. Secondly, we consider that our new setup, which enables the application of biaxial tensile stresses at cryogenic temperatures, is unique, and goes well beyond the uniaxial machines, with its ability to apply two perpendicular tensile stresses on a single sample in a reversible way. This can be used to explore a very large region of a new 3D phase diagram by varying Temperature and tensile strain in two perpendicular directions.

All these new elements have enabled us to go much further in our understanding of this physics. Our work demonstrates the nature of the new CDW along the a-axis, the direct correlation between structure and CDW transition, and that the link between gap and T_c does not follow BCS equation, which questions the theoretical description of CDW in these layered systems. More precisely, in this paper:

- We show for the first time that new CDW satellite reflections appear in XRD experiments along the a-axis while the ones along the c-axis disappear. This directly proves the assumptions found in the literature concerning the existence of a CDW along the a-axis of RTe_3 crystals when they undergo uniaxial deformation.
- We prove that the transition from CDW_c to CDW_a is driven by the a/c structural parameter and takes place exactly at $a = c$. The direct dependence of the transition on a/c was never mentioned in any other study, and the $a = c$ value of the transition never measured before. To do so, we precisely measured 3 non-collinear Bragg reflections to retrieve the exact deformation of the unit cell. This had never been reported before in RTe_3 systems under deformation and was possible because our setup allows an unprecedented combination of experimental techniques under tensile stress and at low temperatures: XRD to follow the atomic and CDW structure, and transport to follow the CDW transition, performed on the SAME sample.
- We prove that the data obtained by XRD and transport are absolutely identical, and that the disappearance of CDW_c and appearance of CDW_a upon increase of a/c are seen both on the atomic and electronic structures.
- We explained in a clear and simple way, comprehensible to a general reader, why the CDW wave vector chooses the direction of the longest lattice constant.
- We demonstrate that T_c evolves linearly with a/c , and provide theoretical support for these, while other quantities, mostly related to the gap, saturate. This had never been shown before, and questions the very nature of the CDW in these layered materials. The fact that XRD and transport measurements saturate while T_c diverges linearly up to the maximum deformations reached here is a puzzling result. The evolution under higher deformations should be explored to find where T_c stops diverging. But in a more general way we believe our results will activate new experimental and theoretical works to understand this behaviour.
- The coexistence region corresponds to square Te planes, and this opens exciting new investigations in this state. We are here able to control the structure finely and reach a stable deformation state, and thus finely tune the electronic state, that can persist as long as the deformation is preserved.
- We perform for the first time equibiaxial deformation as a function of temperature, which was never reported before and highly strengthen the conclusions of our work that the relevant parameter driving the transition is a/c .
- These first results obtained in this cryogenic biaxial tensile stress device will for sure be a starting point for many other studies in condensed matter systems. However, this manuscript is not intended to show results obtained in different condensed matter systems, as it would be too broad topic.

Concerning the development of the new device used in these experiments, Reviewer #2 is right to suggest that we should stress more the importance of this development for these measurements. Our original goal in this paper was to focus on the new physical results obtained on the CDW system TbTe_3 . However, although our aim is still to publish an instrumental article focused on the device itself and its performances, it is also important for the reader to have more information in this manuscript. This is why we replaced reference [10] that mentioned a paper to come, with a reference to an additional section of the Supplementary Materials in which we provide a description of the device and its performances, that are useful for the understanding of the experimental data shown in the manuscript (see section I of the Supplementary Materials).

There is a lengthy discussion connecting the diffraction intensity and electronic gap. While the discussion is likely true, there are no direct measurements of the gap. There is discussion of ARPES measurements to confirm the behavior, but the discussion is speculative and should be acknowledged as such. To make a connection between the current results and the electronic structure, the authors point to Ref. 23, which looks at LaTe3. However, there are other references, e.g. Science 321, 1649 (2008) and Phys. Rev. B 93, 024304 (2016), that investigates TbTe3 under femtosecond laser excitation. It seems more direct connections between the lattice and electronic structures, and the deformation potential could be made by comparing with results from laser excitations on the same material.

Diffraction peaks – including CDW satellites – probe the atomic structure. In the case of CDW reflections, taking into account a periodic lattice distortion, it can easily be shown that the scattered amplitude is proportional to the amplitude of the PLD (provided this one is very small compared to the lattice period). In displacive CDW models, the gap is also proportional to the amplitude of the PLD, which means that the scattered x-ray intensity is proportional to the square of the gap. In RTe3 systems Ru et al. showed that the satellite intensity is proportional to the square of the order parameter (Ru et al, PRB 77, 035114 2008). In our case, we also relate the variations of CDW satellite intensities to gap variations.

Resistivity measurements also display a similar behaviour, and the jump in the resistivities must also depend on the gap, although the exact dependence is not straightforward in general. Here we show that XRD and resistivity data display exactly the same behaviour, which is a strong indication that resistivity measurements provide a good estimation of the gap variations in the present case.

Reviewer #2 is absolutely right that this should be confirmed by direct measurements of the gap. However, performing ARPES measurements under mechanical deformation is not straightforward and requires a dedicated effort. As underlined by Reviewer #2, the two papers related to TbTe3 [Science 321, 1649 (2008) and Phys. Rev. B 93, 024304 (2016)] provide separate results on tr-ARPES on the one hand and on tr-XRD on the other hand, which makes totally sense. The data shown in our manuscript are consistent, reproducible and already sufficiently complete to allow us to reach our conclusions.

Concerning out-of-equilibrium measurements, we refer to Kogar et al. [Ref 23] because their results are directly related to what we observe, i.e. additional satellites along the a-axis. The two references indicated by Reviewer #2 [Science 321, 1649 (2008) and Phys. Rev. B 93, 024304 (2016)] are indeed related to TbTe3, i.e. the same system as the one of our manuscript, and explore transient states after laser excitation. The correlation between the two very nice results reported in the two papers demonstrate that XRD and ARPES measurements probe the same amplitude mode of the CDW, and confirm that XRD is a good probe of the gap in TbTe3. This has been added in the text to strengthen the link between gap and XRD data in the manuscript, together with the two references.

On a clerical note, the order of the references should be checked, e.g., I see a pointer to Ref. 12 before Ref. 11 in the main text.

Ref. 11 (now Ref. 10 in the revised version) is in the legend of Fig. 1, that is meant to be read before the appearance of Ref. 12 in the read order (now Ref. 11 in the revised version). We also cite Ref 10 before now.

Reviewer #3 (Remarks to the Author):

Gallo-Frantz et al. present a comprehensive study of stress-tuned charge density wave (CDW) transitions in rare earth tritellurides (RTe₃) using novel techniques involving x-ray diffraction and transport measurements under bi-axial stress at cryogenic temperatures. The paper is well-structured and convincingly presents a set of new data revealing the behavior of CDW in RTe₃ under both symmetry breaking ($F_a \neq F_c$) and symmetry preserving ($F_a = F_c$) bi-axial stress. This paper is suitable for publication after the authors address the following questions and suggestions.

The authors would like to acknowledge Reviewer #3 for his/her careful reading of the manuscript and his/her relevant comments. Our answers to his/her questions are listed below.

1. Role of Lattice Parameters: In Figure 3(d-f), the authors observed that the CDW state and transition temperature are barely affected by symmetry preserving biaxial stress. This observation contrasts with previous reports, such as PhysRevB.77.035114, which found that the CDW transition changes with chemical pressure. If the in-plane a and c lattice parameters do not play a key role, as suggested by the authors, can the authors provide their insights and discussions on what instead determines the CDW transition temperature? For example, as the symmetry preserving biaxial stress is applied, how does the out-of-plane lattice parameter b change?

Hydrostatic pressure and chemical pressure experiments give a consistent picture of gap and transition temperature variations.

Our experiments explore the effect of in-plane tensile stresses and give rise to different behaviours. In our case, the structural changes are intrinsically very different from the ones performed under hydrostatic or chemical pressure. Both chemical and hydrostatic pressure experiments result in a change of all lattice parameters in the same direction (they all increase or decrease simultaneously), and thus in large volume changes. With chemical pressure, from La to Tm, the unit cell volume changes by nearly 10%. In our case, when applying in-plane tensile stresses, either uniaxial or biaxial, the out-of-plane lattice parameter b decreases while one or both in-plane lattice parameters increase. The total volume does not vary much: in the uniaxial experiment presented in the manuscript, the maximum volume change is 0.3%.

The equibiaxial deformation method, although preserving the in-plane symmetry, leads to a decrease of the b parameter. We unfortunately could not get the lattice parameters of the specific sample presented in the manuscript, but experiments performed in other sample could demonstrate this. The fact that the out-of-plane lattice parameter decreases with equibiaxial in-plane stresses also contrasts with the structural changes induced by chemical pressure. The role of the out-of-plane lattice parameter should be further explored to draw any conclusion on this point.

Understanding why the transition temperature evolves in a different way here than in these two other kinds of pressure-induced experiments is not straightforward at all. It would require some additional theoretical effort, both analytical and numerical. If the model provided in the text describes the relative variation of T_c induced by a relative change of lattice parameters, the determination of the exact value of T_c as a function of rare-earth element and lattice parameters goes beyond this. T_c is determined by the balance between electron energy gain (due to the gap opening) and entropy (due to disorder), but RTe₃ systems have different gap values and electron densities near Fermi level, as well as different $t_{//}$ and t_{\perp} transfer integrals, which induce different conditions on the Lindhard susceptibility and Coulomb interactions, and thus different intrinsic T_c , even if $a = c$. In all cases, we only address here the relative variations of T_c due to strain in TbTe₃.

2. Clarification of CDW Domains: In lines 247 to 250, the authors concluded that they did not observe any change in the width of the Bragg peaks, which does not support the appearance of domains during the transition from CDW_c to CDW_a. However, in preprint arXiv:2306.14755, Fig. 2(c), a clear discontinuity in the a-axis lattice parameter was observed, suggesting the coexistence of CDW_a and CDW_c domains. Additionally, the width of Bragg peaks in the preprint displayed a finite dependence on strain. Can the authors comment on the discrepancy between their results and those presented in arXiv:2306.14755?

The width of the CDW peaks obtained in the rocking scans indeed do not change within our resolution. The width of rocking scan peaks is determined by several factors: incident beam divergence, crystal (or CDW) mosaicity, and domain size. Here, the beam divergence and mosaicity dominate the peak widths, but domains smaller than 25nm should induce a broadening of CDW satellite peaks. We thus agree that if domains appear with a size larger than 25 nm then our experiment could not resolve it. We modified the text to take this value into account. However, other transport measurements we performed independently from the data shown here also support the assumption that the formation of CDW domains is unlikely during the transition. This will be further studied. Besides, the data shown in Fig. 2(a) of arXiv:2306.14755 seem to show that the CDW peaks are self-similar with applied stresses (although largely spread in reciprocal space), and only the total intensity seems to vary, which is also in agreement with a pure CDW amplitude variation as a function of applied stresses.

The diffraction data presented in Fig.2(c) of preprint arXiv:2306.14755, and mentioned by Reviewer #3, show the evolution of the 2 20 0 lattice Bragg peak as a function of applied uniaxial stress along a. First it seems there is a discrepancy in the direction of CDW mentioned in the legend, a-axis and c-axis CDW are inverted with respect to Fig. 2(a) and 2(b). It is difficult for us to discuss these data because the way the information is extracted is not clear to us. First, from only one Bragg reflection, here the 2 20 0, it is not possible to compute the value of the in-plane lattice parameter a without any assumption on the out-of-plane lattice parameter b . We know from our measurements that the latter decreases with applied stresses, and this should be taken into account for the determination of a . The robust way to compute lattice parameters is to measure 3 non-collinear Bragg reflections. This would help to check that the c lattice parameter shows the same behaviour. Moreover, the data show a projection of the XRD data along a , meaning that this cut is performed along the a^* reciprocal direction, i.e. the longitudinal direction for CDW_a. The methods to do this is not explained in the text, so it is difficult for us to appreciate if the projection is correct. For instance, if these data were obtained by converting the angles of a rocking scan into lattice parameters, then they correspond to transverse direction and cannot be interpreted as two superimposed lattice parameters. It should also be checked in these measurements how the experiment was performed to eliminate any hysteretic behaviour between compressive and tensile stresses. In all cases, seeing that the Bragg reflection splits at the transition is not a proof of the formation of CDW domains. The information on CDW domain formation has to be sought on the CDW reflection. To conclude on the data shown in the preprint mentioned by Reviewer #3, although likely correct, these measurements on the 2 20 0 Bragg reflection should be clarified to draw any conclusion on the possibility to observe CDW domains on the Bragg reflection itself. To our point of view, there is no discrepancy between this preprint and our manuscript: the data obtained on the CDW reflections in both studies are consistent.

In any case, the answer to the presence of domains or not should be the scope of additional dedicated measurements, and will be performed in future experiments.

3. Symmetry of TCDW vs. a/c : Due to the existence of a glide plane, the system does not retain higher symmetry even at $a = c$. Therefore, there is no apparent reason for TCDW vs. a/c to be symmetric with respect to $a = c$. Nevertheless, figure 4(c) demonstrates that TCDW has a linear dependency on a/c and is almost symmetric with respect to $a = c$. The author's argument is that the glide plane plays no role in the CDW formation. However, x-ray diffraction data in Fig. 4(a) shows that the CDW_a and CDW_c states saturate to different maximum intensities, indicating that they are not identical, hence the CDW intensity is asymmetric with respect to $a = c$. The authors should provide an explanation for the inconsistency between these two sets of data.

This discussion addresses the main point of our manuscript : what is the link between gap and T_c in this system ?

T_c is indeed symmetrical with respect to $a = c$, meaning that the only parameter that matters for its variation is a/c , as indicated in the manuscript. We can therefore conclude from this observation that the glide plane does not seem to play a crucial role in the observed relative variation of T_c , and that the transition from CDW_c to CDW_a is related to the symmetry of the Te sheets, with a transition when they display a square 2D lattice.

As Reviewer #3 points out, and as discussed in the manuscript, the intensities of the satellites related to CDW_a and CDW_c do not saturate to the same values when $a/c \gg 1$ and $a/c \ll 1$ respectively. First, this measurement is robust: we also observe the same in transport measurements, with good quantitative agreement, and have measured this in several samples (as shown in the Supplementary Material). From our measurements, we can estimate that the gap obtained in the CDW_a phase is 80% the one of CDW_c (if the satellite intensity is related to the square of the gap). This behaviour - and the exact value of the gap - has to be measured by other means in future experiments. But indeed, this means that, contrary to T_c , the gap is not symmetric along a and c , and that it is not only fixed by the symmetry of the Te planes. The glide plane could play a role in this, but we cannot guarantee this.

For example, if the wavevectors obtained in the saturation regime of CDW_a and CDW_c are different, then the coupling constants will also differ along a and c (Eq. 4 in the manuscript). The CDW amplitudes (and thus gaps) are related to these coupling constants. In the case of different CDW wavevectors along a and c , we will thus obtain different saturation values for the respective gaps. This should appear in high-resolution XRD experiments.

However, even if the CDW wavevectors are symmetric along a and c , the gap values still should differ. The CDW wave vector and TCDW are mainly determined by the maximum of the product of susceptibility and CDW coupling (see Eq. (2)). Substituting Eqs. (3) and (4) to Eq. (2) we see that this product is symmetric to the exchange of a and c , and so does T_c . The CDW gap value is determined by the overall energy gain, which also depends on the elastic constants. Here the asymmetry between a and c is already seen from Fig. 2b of the manuscript, where at the same applied stress the lattice constants differ along a and c , which is also determined by the energy gain. The glide-plane asymmetry could contribute to this asymmetry of elastic constants.

In all cases, the aim of this manuscript is first to provide a first robust experimental observation, and we are sure that it will generate further studies, from our group or other ones, both experimentally and theoretically, to understand the nature of these CDWs in RTe₃ systems. Our data are consistent and reproducible, and we have to go beyond to provide a clear picture of the mechanisms.

4. Tight-Binding Model: the authors used a tight-binding model to explain the linear dependence of TCDW on a/c . Does the model also predict the lack of change in TCDW under biaxial stress? If not, the authors should elaborate on why the tight-binding model is not applicable in the condition of biaxial stress.

The tight binding model we use successfully accounts for the observed variations of T_c in the uniaxial experiments, starting from a theoretical T_c defined at $a = c$. The observed linear behaviour is well described, with good quantitative agreement. The question of the equibiaxial experiment can also be addressed within this model.

As we noted in the text, there are several effects of stress on the CDW transition temperature TCDW. The main effect is the dependence of the CDW coupling on the CDW wave vector, described by Eq. (4) in our paper. The second effect is the change of the electron density, which results to the shift (and maybe additional curvature) of Fermi surface, and, hence, to the change of CDW wave vector, the electron susceptibility and the transition temperature T_{CDW} . The third effect is the change of the lattice constants themselves. It does not directly affect the susceptibility given by Eq. (3), because it contains the sum over quantum states, which is equivalent to the integrations over the products ak_x and ck_z . However, the change of the lattice constants produces the slight change of the transfer integrals $t_{//}$ and t_{\perp} , which affects the susceptibility (for example, via the Fermi-surface nesting property). The biaxial stress also affects the CDW transition temperature via these three mechanisms, ordered corresponding to their importance. But their effect for biaxial stress is smaller than for uniaxial.

The main effect comes from the first mechanism, which is much stronger for uniaxial stress. During the uniaxial stress the lattice constants change oppositely, the unit cell volume and the electron density do not change in the zeroth-order approximation, as this would cost much energy. Hence, the change of the lattice constant along the stress is stronger for the uniaxial than for the biaxial stress of the same magnitude. The change of CDW wave vector, of CDW coupling according to Eq. (4) and of the CDW transition temperature are thus also stronger for uniaxial than for biaxial stress. In addition, the second and the third effects described above, partially, compensate the first effect for the biaxial stress but not for uniaxial.

5. Technical Details: Is the experimental setup capable of applying compression stress? If not, why?

We added a section in the Supplementary Materials in which we provide more information on the new device developed for these experiments. As the sample is glued on a deformable substrate, only tensile stress can be applied. However, when applying uniaxial tensile deformation in a direction, the two perpendicular ones are in compression due to Poisson ratio, so in all cases, all deformation components have to be measured (by XRD on 3 non-collinear Bragg reflections here). But formally, no in-plane compressive stress can be applied as the deformable substrate cannot transfer it.

Reviewer #4 (Remarks to the Author):

The authors would like to thank Reviewer #4 for the time he spent on this work.

REVIEWERS' COMMENTS

Reviewer #1 (Remarks to the Author):

The authors have addressed the concerns raised in the original manuscript, and it is suitable for publication now.

Reviewer #2 (Remarks to the Author):

The authors have carefully considered the comments and concerns of all the reviewers. The rebuttal and alterations to the manuscript are satisfactory and I recommend the manuscript for publication.

Reviewer #3 (Remarks to the Author):

the authors have addressed the questions and concerns adequately. The paper is now suitable for publication.

Reviewer #4 (Remarks to the Author):
